# The Coming Emptiness: On the Meaning of the Emptiness of the Universe in Natural Philosophy

## Gregor Schiemann

Philosophical Seminar and Interdisciplinary Centre for Science and Technology Studies (IZWT), University of Wuppertal, Gaußstr. 20, D-42119 Wuppertal, Germany; schiemann@uni-wuppertal.de

**Abstract:** The cosmological relevance of emptiness—that is, space without bodies—is not yet sufficiently appreciated in natural philosophy. This paper addresses two aspects of cosmic emptiness from the perspective of natural philosophy: the distances to the stars in the closer cosmic environment and the expansion of space as a result of the accelerated expansion of the universe. Both aspects will be discussed from both a historical and a systematic perspective. Emptiness can be interpreted as "coming" in a two-fold sense: whereas in the past, knowledge of emptiness, as it were, came to human beings, in the future, it is coming, insofar as its relevance in the cosmos will increase. The longer and more closely emptiness was studied since the beginning of modernity, the larger became the spaces over which it was found to extend. From a systematic perspective, I will show with regard to the closer cosmic environment that the Earth may be separated from the perhaps habitable planets of other stars by an emptiness that is inimical to life and cannot be traversed by humans. This assumption is a result of the discussion of the constraints and possibilities of interstellar space travel as defined by the known natural laws and technical means. With the accelerated expansion of the universe, the distances to other galaxies (outside of the so-called Local Group) are increasing. According to the current standard model of cosmology and assuming that the acceleration will remain constant, in the distant future, this expansion will lead first to a substantial change in the epistemic conditions of cosmological knowledge and finally to the completion of the cosmic emptiness and of its relevance, respectively. Imagining the postulated completely empty last state leads human thought to the very limits of what is conceivable.

**Keywords:** natural philosophy; cosmology; emptiness; vacuum; void; dark energy; space flight; exoplanet; big freeze; big crunch; everyday lifeworld

## 1. Introduction

The cosmological relevance of the emptiness of the universe is not yet sufficiently appreciated in natural philosophy.[1] By the emptiness of the universe or cosmic emptiness—or, for short, emptiness—in what follows, it will be understood as the space in the universe that is free of bodies. Bodies occupy limited and isolated three-dimensional regions of space and consist of a quantity of matter in an aggregate state with values above a minimum density [2]. Typical examples of bodies are asteroids, moons, planets, and suns. The definition of a body excludes interplanetary, interstellar, and intergalactic media consisting of dust, gases, and low-density plasmas. On account of their low density, these media are almost like a vacuum, which as matter-free space, belongs to emptiness.

---

[1] The universe is observable space, and its rational description is called the cosmos. Because observation is rational description, I use the terms "universe" and "cosmos" synonymously. See also Kragh [1] (p. 1 ff.).

The concept of a vacuum is ambiguous. Substances of low density are also referred to as vacuums [3] (p. 236). Thus, there is a shifting boundary between a vacuum and emptiness.

Emptiness can be interpreted in natural philosophy as "coming" in a two-fold sense: whereas in the past, knowledge of emptiness came to human beings, as it were, in the future, it is coming, insofar as its relevance in the cosmos is going to increase. In what follows, the space in the universe without bodies will be discussed from the perspective of natural philosophy, whose subject is nature, knowledge about nature, and human beings' relationship to it.[2]

Natural philosophy is, methodologically speaking, part of philosophy and has a systematic, historical, and practical character [5]. With regard to the cosmos, it inquires into its structures and the place of human beings in it, but also into the history of cosmological knowledge and its practical contexts. It has the same subject matter as the natural sciences, relies on their results, and shares with the philosophy of science an interest in the conceptual foundations and formal structures of scientific theories. However, its central point of reference remains the significance of nature for human beings. From a natural philosophical perspective, defining emptiness as body-free space recommends itself for two reasons. Firstly, natural philosophy, which concerns human beings' relationship to nature, should also be comprehensible for nonscientists. Bodies are tangible objects insofar as they are accessible to sensory perception, which is usually the case. Secondly, body-free space is characterized by a special, namely, a hostile relationship to human life.[3] I will defend the theses that in the past, the discovery of the cosmic emptiness occurred in a cumulative way and that the future living conditions of civilizations in the cosmos will be increasingly determined by their relationship to emptiness.

The theme of cosmic emptiness was and is closely linked to the concept of nothingness in natural philosophy, despite the controversial nature of this connection. In part, only space devoid of matter is called nothingness, and in part, the more inclusive concept of emptiness is used; sometimes, conversely, nothingness is contrasted with everything that exists, which also includes space devoid of matter.[4] In what follows, I will not examine the relationship between nothingness and cosmic emptiness more closely. However, I would like to preface my remarks by saying that they are essentially motivated by the conviction that today, the philosophical concept of nothingness cannot be understood without reference to the knowledge of cosmic emptiness.[5]

Today, cosmological knowledge is based on highly technical observational instruments and on exceedingly abstract mathematical theories and models. Its object domain extends from the tiniest spatial dimensions on Earth to the edge of the observable universe, which is about 45 billion light years away, and from the briefest times at the claimed beginning of the universe to the most distant future times of the claimed final state of the expanding universe. For natural philosophy, which inquiries into the relationship between human beings and the cosmos, and thus also into the significance of cosmological knowledge for the human understanding of the world, cosmological knowledge represents a challenge. How can nonscientists incorporate the results of this demanding and unintuitive knowledge into their conception of nature? That there are some starting points for this is shown by the widespread interest in popular accounts of cosmological knowledge. Insofar as the cosmological questions in natural philosophy refer to future possibilities for action and states of

---

[2]    Insofar as "nature" is understood as that which exists in itself without being produced by human beings, emptiness is a natural phenomenon. On the concept of nature, see Schiemann [4].

[3]    In highlighting its hostility to life, the natural philosophical thematization of cosmic emptiness differs from that of cosmology. From a cosmological perspective, emptiness can be regarded as a condition for the emergence of possibilities of life in the cosmos, insofar as a greater density of matter in the universe would have prevented this emergence.

[4]    The definition of nothingness as space devoid of matter (a vacuum) is widespread; see, for example, Genz [6], Close [7], and Krauss [8]. The epitome of cosmic emptiness is cosmic spaces of very low density (voids), which are also called nothingness; see, for example, Sheth and van de Weygaert [9] and Szapudi [10]. For the contradictory juxtaposition of nothingness and being, including space devoid of matter, see, for example, Albert [11].

[5]    The cosmic dimensions of emptiness are completely absent from definitions of nothingness in contemporary philosophy.

the universe, the natural philosophy of cosmology has a speculative character that also represents a challenge to its scientific character.

The assumption that modern cosmology can make predictions extending into the most distant future calls for an extended concept of natural philosophy, whose scope encompasses not only existing human beings. Cosmology deals with future times, concerning which we cannot know whether human beings or extraterrestrial civilizations will still exist, and with even more distant times, in which it is now assumed that life will no longer exist in the universe. Therefore, the concept of natural philosophy must also refer to the merely possible existence of humans and extraterrestrial civilizations as well as to a universe without thinking beings. Such a concept of natural philosophy is without historical precursors. Traditional natural philosophy denies the truth of statements that refer to states of distant past or future times in which subjects of cognition did not yet or no longer exist.[6]

I will concentrate on two aspects of cosmic emptiness that I regard as especially relevant: the distances to the stars in the closer cosmic vicinity of the Solar System and the accelerated expansion of the universe. Between the Sun and the stars, there lies an emptiness that is hostile to life and may not be traversable by humans, the accelerated expansion will probably empty the universe completely of bodies. I will leave other natural philosophical questions concerning cosmic emptiness aside, such as the problems of interplanetary space travel or the importance of the cosmic emptiness for the evolution of the cosmos and its habitable places, which would probably not exist without the cosmic emptiness. The two aspects selected are suitable for thematizing the cosmic emptiness from the closest cosmic environments to their most distant future states. These aspects are among the subjects that in recent times have experienced a stormy development in knowledge marked by paradigm shifts and are currently at the center of cosmological interest. This makes it all the stranger that to date, they have received scarcely any attention in natural philosophy.[7]

Part of the appeal of these two aspects is also their antithetical character with respect to the philosophy of nature. The distances between the stars in the closer cosmic vicinity of the Solar System can be discussed with direct reference to everyday knowledge. They are visible to the naked eye and their distance from the Earth can be translated into travel times measurable in human lifetimes. However, the accelerated expansion of the universe—assuming that it remains constant—leads into distant cosmic spaces and times that not only lie far beyond all everyday practical experience, but also extend to the limits of what is conceivable. At a time that is assumed to begin, at the earliest, about $10^{100}$ years from now (a period that can no longer be expressed in common words),[8] accelerated expansion will have led to the dissolution of all bodies. Thus, this aspect includes a period of time in which life will no longer exist. A consideration of the current epistemological conditions of cosmological knowledge will show that these distant scenarios can also be relevant for the human understanding of the world.

I will begin the first part with a brief outline of the history of the discovery of interstellar and intergalactic distances in order to show that the historical development of the knowledge of this aspect of cosmic emptiness was indeed cumulative and thus approached human beings, as it were. The closer cosmic environment will be localized within the observable universe as a vanishingly small section. Then, I will go on to discuss conditions of the future relevance of the cosmic emptiness of interstellar space for natural philosophy. Here, the focus will be on exoplanet research. In the second part, I will turn to the expansion of the universe. Again, I will begin its natural philosophical interpretation with a short summary of the cumulative character of the history of its discovery. In the third part, I will go on to discuss two future cosmic events resulting from the claimed accelerated expansion, namely,

---

[6]  In this, I am following Quentin Meillassoux's and Ray Brassier's criticism of natural philosophy in the tradition of Immanuel Kant. Brassier has applied Meillassoux's justification of the truth of prehistoric (ancestral) statements to post-historical (posterior) statements [12] (p. 229 f.).

[7]  Reflections on emptiness in natural philosophy are limited to the concept of the vacuum as space devoid of matter and can be found in popular accounts by physicists (e.g., [6–8]) and in discussions in the philosophy of science (e.g., [13,14]).

[8]  The character "^" is placed before power numbers instead of writing them as superscripts.

the substantial change in the cognitive conditions of cosmological knowledge that they will bring about in the distant future and the completion of emptiness that will be brought about in the final state of the universe. In the concluding fourth part, I will discuss the objectivity and the current relevance of the cosmic emptiness.

## 2. Cosmic Distances and Interstellar Emptiness

### 2.1. The Emerging Knowledge

Until the beginning of modernity, cosmological conceptions of the relations of distance between the stars in the universe were shaped by the ancient notion of the closed celestial sphere. According to this notion, the stars were situated at the inner edge of a closed sphere encircling the Earth. The assumption that this sphere might not exist and that the stars might be arbitrarily far away from the Earth is characteristic of the modern revolution of cosmological ideas. The decisive impulse for this revolution was provided by the foundation of the heliocentric worldview by Nicholas Copernicus. Reflections in astronomy and the theory of gravitation subsequently supported the possibility of an infinite universe.[9] Blaise Pascal recognized that the presumed infinity of space is associated with a hostile emptiness when he wrote: "The eternal silence of these infinite spaces frightens me." [17] (p. 36).

It was not until the seventeenth century that improvements in measurement techniques made it possible to determine the distance to the Sun and thus to the planets. The values for the distance to the Sun exceeded the relevant assumptions of the medieval worldview by a factor of about 20 [1] (p. 26).[10] In the nineteenth century, it was also the advances in measurement techniques that enabled Friedrich Wilhelm Bessel to determine the distance to a nearby fixed star for the first time. The value of ten light years shocked the educated world. Edgar Allan Poe wrote with regard to Bessel's results of a "terrible gap" that separates the suns from each other and of an "immeasurable distance" that cannot be bridged by human beings [18] (p. 85 f.).

Since the technical possibilities of observation extended essentially only to the stars of the Milky Way and it was not possible to identify the neighboring galaxies as such without doubt, the vast majority of astronomers believed until the beginning of the last century that the stars of the universe were confined to the Milky Way. In 1925, Edwin Hubble, using the largest reflecting telescope at the time, succeeded in providing the groundbreaking first evidence of a galaxy (the Andromeda Nebula) outside the Milky Way, whose estimated distance of around one million light years was too small by about half, but nevertheless exceeded by far the previously assumed orders of magnitude for the Milky Way [19] (pp. 509 f.), [1] (p. 119).

Today, it is estimated that there are around two hundred billion galaxies in the observable universe, with the most distant galaxy being located at a distance of about 32 billion light years from the Earth [20,21]. The arrangement of galaxies probably exhibits a honeycomb-like structure, whose cavities have the lowest matter density in the universe and constitute its largest structures. At the provisional end of the discoveries of ever-greater distances between the cosmic bodies is the proof of the existence of these voids, whose diameters of up to one billion light years (of the Eridanus Supervoid) eclipse all previous notions about the size of body-free spaces [22].[11] In the regions of space close to the Sun,

---

[9] If the sun and not the Earth is located at the center of the universe, it must be possible to measure the distances to not infinitely distant stars from the parallactic shift of their measurable positions. Since this shift could not be demonstrated (on account of deficiencies of measurement techniques), Giordano Bruno assumed that the stars were infinitely distant [15] (p. 11). According to Isaac Newton, gravitational attraction would inevitably lead to the collapse of the universe if it were not infinite [1] (p. 73).

[10] On the amazement provoked in his contemporaries by Giovanni Cassini's first measurement of the distance between the Earth and the Sun in 1672, see Ferguson [16] (pp. 136 f.).

[11] Measurements or estimates of cosmic distances can be regarded as certain for very large distances only since the end of the last century.

measurements of distances remain on the same scale as Bessel's discovery. The closest star is four light years away, and within a radius of 10 light years, there are just 15 more stars [23].

From the cumulative history of the discovery of cosmic distances, natural philosophy derives the insight that with every additional step in the disclosure of the scale of the cosmic emptiness, the space occupied by the Earth in relation to it diminishes. Spatially speaking, the Earth has, as a result, shrunk to almost nothing. The fact that all distances are measured in a single unit and that there is a gradual transition from the smallest distances to the largest means that the proportions of the cosmos can be illustrated by reducing the scale. If you reduce, for example, one million light years to one millimeter, you get a handy model of the universe of about 90 m diameter, in which, however, the Milky Way, with about 0.2 mm diameter, is already almost invisible. These analogies, with which the observer reduces him- or herself to nothing, form the basis for also rendering very large spatial magnitudes imaginable in terms intelligible from the lifeworld.

From the perspective of natural philosophy, however, there are also categorical differences between the spatial orders of magnitude. They can be explained in terms of the distance-dependent dominance of natural forces: within very small dimensions, different forces operate than in very large dimensions. In subatomic orders of magnitude, for example, the weak and strong nuclear forces are determinant, but not the gravitational forces that hold exclusive sway on the very large scale. In what follows, I will present a different, likewise natural philosophical explanation for the categorical differences between the spatial orders of magnitude, one based on the presumed limited possibilities for human action in cosmic space.

### 2.2. The Proximity of Emptiness

We can gain an initial impression of human beings' current scope for action in space by translating distances into the travel times of today's spacecraft. Based on the order of magnitude of the speeds for the Moon flights (10 km/s), all of the planets in the Solar System can be reached within travel times that still allow astronauts to return to Earth during their lifetimes. However, the outward journey to a star 12 light years away, that is, one in the immediate cosmic vicinity of the Solar System, would already take over 350,000 years.[12] This would justify a categorical difference, of course, only if it were not possible for human beings to travel at speeds sufficiently close to the speed of light to reduce travel times for stellar distances to a human scale. Traveling at approximately the speed of light would slow time down for the travelers so much that they could travel cosmically close distances without having to countenance substantial time differences from terrestrial time. Assuming that a spaceship were set in motion and slowed down at rates of acceleration and deceleration such that the passengers were exposed in each case to the same force as the gravitational force on Earth and that it reached a maximum of 99% of the speed of light, the astronauts would be back from a journey to a star 12 light years away (without a stay there) within 28 years, while for them, only 10 years would have passed.

In order to discuss the room for realizing such journeys allowed by natural laws, all technical difficulties will be set aside in what follows—for example, the problem of manufacturing and transporting suitable fuels or the problem of high-speed collisions with small particles located in interstellar space. A thought experiment for calculating an ideal rocket engine stems from the Nobel Prize winner Edward Mills Purcell [24] (pp. 6–9). In order to convert the fuel mass completely into energy, matter and antimatter would have to be brought together. Since in that case, the energy would be released in the form of light particles (photons), the speed of the emerging beam corresponds exactly to that of light. The relativistic rocket equation for this spaceship on a return flight to a destination 12 light years away with a maximum speed of 99% of the speed of light and a double constant acceleration and deceleration process still yields results involving a seemingly absurd ratio between the initial mass, when no fuel has yet been consumed, and the final mass. To transport a

---

[12]   It would take over 100,000 years to reach the closest star four light years away.

payload of ten tons, a spaceship with a total mass of at least 400,000 tons would have to be built [24] (p. 8). The power required to accelerate and decelerate such a spacecraft would be the equivalent of almost 60,000 times the current total annual primary energy consumption of the Earth.[13]

With this calculation, Purcell intended to show that plans for interstellar journeys with missiles are childish notions that should not be taken seriously. In the light of such criticism, physicists and technicians contemplating future space vehicles for interstellar journeys have been looking for forms of propulsion that make do without rocket technology.[14] Vehicles driven by external laser beams can be regarded as an example for a realistic option. The light source would be situated at a fixed location and would drive a spaceship even at great distances using the radiation pressure of light. With this technology, however, only speeds that remain significantly below the speed of light can be achieved. They require a power output that can only be generated for very low payloads on Earth, but otherwise would also correspond to many times the annual primary energy consumption of the Earth.[15]

There is no known physical theory according to which it would be impossible in principle to produce the physical power required to achieve speeds close to the speed of light. Given the current state of knowledge, however, there are valid reasons for doubting whether the necessary outputs of power can be generated in the future. Without relativistic velocities, the travel times of interstellar manned space travel must be assumed to be many times longer than the average current lifespan of a human being. In order for people to be able to live in complete isolation from the outside world over several generations, Earth-like conditions would have to prevail in the spaceships (intergenerational spaceships).

When it comes to human beings' scope for action, therefore, there are good reasons to assume a categorical difference between the interplanetary distances of the Solar System and the far greater interstellar distances. From a cosmic perspective, human beings' possibilities for action, taking into account their technical possibilities, are characterized by a connectedness to the Earth that essentially limits the spatial range of actions in the cosmos to the Solar System for the time being. However, this entails a further difference that is characteristic of the relationship between human beings and the cosmic emptiness: human beings' possibilities for acting on a cosmic scale are in clear contrast to their observational capabilities, which, as we have seen, extend to the limits of the observable universe.

I regard exoplanet research as a candidate for a paradigmatic case of this contrast. The successes of this research can be regarded as outstanding achievements of a technically perfected precision astronomy, whose beginnings can be traced back to the discovery of a planet in a different Solar System for the first time in 1995. Since then, over 3700 planets have been identified [30]. In the meantime, it is considered probable that suns normally have planets. Exoplanet research is a form of observation-based research that deduces its findings indirectly from the analysis of the radiation of the suns around which the exoplanets orbit. From changes in the stellar data, it infers not only to the existence of their planets, but also, where possible, to the extent of their mass, their volume, and their distance from the sun in question, as well as to the existence and the composition of the atmosphere. The precision of the measurements already allows researchers to study whether the planets in question are possibly

---

[13]   The power consumption of the spaceship was calculated following Purcell at $1.14 \times 10^{18}$ watts [24] (p. 8); the annual primary energy consumption on Earth, at $1.73 \times 10^{13}$ watts, was derived from the data for 2014 provided by the German Federal Agency for Civic Education [25].

[14]   Matloff [26] and Long [27] provide relevant introductions to alternative rocket propulsion systems.

[15]   The Breakthrough Starshot project plans to bring a very low payload of just a few grams to the closest star, Alpha Centauri, at about 4 light years away, with an Earth-based laser. Assuming a laser power of 10ˆ11 watts, with which it is planned to drive the spaceship from Earth, the hope is to reach 15–20% of the speed of light [28]. Ian A. Crawford calculates that in order to bring a spaceship with a total mass of 1920 t (with the payload accounting for 450 t) to Alpha Centauri in 36 years, a power expenditure of approx. $10^{14}$ watts would be required for an external laser drive in order to reach a cruising speed of 12% light speed after a period of acceleration of $0.35 \text{ m/s}^2$ lasting approx. three years [29] (pp. 388 f.). This represents over 10 times the annual primary energy consumption of the Earth (cf. [25]), and with current technology, could only be generated in space with sunlight.

habitable, where habitability refers to life as it is known from the Earth. Since very diverse forms of life have developed under terrestrial conditions and in the case of many terrestrial organisms, it is uncertain whether they could develop under other conditions, establishing the criteria for habitability is an extremely complex problem—not least also because it has not been conclusively established even for the Earth which conditions must be fulfilled in order for simple and complex life to be formed.[16] The number of the exoplanets that have been discovered and are considered to be only possibly habitable currently ranges from 10 to 40. The closest of them is already located at our neighboring star 4 light years away, whereas the farthest is almost 3000 light years away [32]. Estimates of the number of possibly habitable planets in the Milky Way are currently of the order of one billion [33].

In future, further improvements in measurement methods and accuracies should make it possible to prove the existence of extraterrestrial (simple and complex) life forms and civilizations, assuming they exist. The existence of extraterrestrial life could become probable by proving that the only way either an observed change in the atmosphere of an exoplanet or a detected electromagnetic signal could conceivably be brought about is by an extraterrestrial civilization [34] (p. 161), [24] (pp. 9 ff.). The general prevalence of planets and the relative frequency of their presumed habitability mean that the probability that extraterrestrial life forms exist has increased significantly. Nevertheless, whatever the findings of further exoplanet research may be, it will be apt to underline the relevance of cosmic emptiness. Proof of the existence of extraterrestrial life forms could render the perhaps unbridgeable distance between the Earth and other stars palpable in a new way. The discovery of the first possible destinations of interstellar communication and travel would merely serve to render their inaccessibility apparent. Conversely, whereas the lack of proof might not yet be sufficient to demonstrate the nonexistence of other forms of life in the closer cosmic vicinity of the Solar System, it could reinforce the impression that an increasing number of exoplanets that are hostile to life merely increases the weight of the emptiness enclosing human beings.[17] Viewed from a natural philosophical perspective, human beings do not evade emptiness. As soon as they lift their (scientific) gaze from the Earth to the cosmos, they are struck by emptiness.

## 3. Distant Emptiness

The result of the discussion of cosmic distances and interstellar emptiness in natural philosophy will be made more evident by the phenomena of accelerated expansion. With the latter, the thematic focus shifts from spatial distances to future times. The knowledge of cosmic distances has rendered the contrast between cosmic and terrestrial dimensions and the difference between cosmic possibilities for action and observation apparent. The phenomena of the accelerated expansion of the universe are located in a future in which Earth's lifespan relative to the cosmic eons that will already have elapsed will be vanishingly small. With this, the predictions of cosmological theories extend into a time beyond all current scopes of action. One of the manifestations of cosmic emptiness discussed—the presumed enclosure of the Earth by a cosmic environment that is hostile to life—will be led to a new future level by the phenomena of the accelerated expansion of the universe, namely, to galaxies and their civilizations, assuming they still exist in the distant future, becoming isolated from the rest of the universe.

For a long time, the accelerated expansion of the universe will have no effect on the stars visible from Earth, even when viewed in cosmic terms. Assuming that it remains constant, the expansion consists of an inexorable enlargement of space that manifests itself in the increase in the distances between galaxies. It will give rise at a distant future time of around 100 billion years from now to a permanent substantial change in the cognitive conditions of cosmological knowledge and will finally

---

16　Cockell et al. [31] provide one of the many overviews of the criteria of habitability.
17　The term "emptiness" could be used in a metaphorical sense to refer to the life-threatening space as a whole and thus include both space devoid of bodies and bodies hostile to life.

lead to the dissolution of all bodies. Whether people originating from the Earth or other civilizations will still exist in 100 billion years' time cannot be known today. Life will probably have ceased to exist in the universe long before the dissolution of bodies. In order even to thematize the accelerated expansion of the universe in natural philosophical terms, we need an expanded conception of natural philosophy that is not limited to the possible presence of human cognition.

*3.1. The Discovery of Accelerated Expansion*

Without disputing its cumulative character, the discovery of the accelerated expansion of the universe can be characterized as a breakthrough in cosmological knowledge in a number of respects.[18] It can be regarded as the final link in a chain of discoveries that refuted the assumption extending back to antiquity that the cosmos was unchangeable. This assumption was originally the temporal counterpart of the already mentioned idea of the closed celestial sphere, but it can be documented historically in related conceptions for a much longer time than the latter—namely, into the first three decades of the last century. It postulates that the size of the universe remains eternally constant [19] (pp. 517 f.). By contrast, the changeability demonstrated by the accelerated expansion of the universe involves an ubiquitous, constantly intensifying dynamic. The energy density attributed to accelerated expansion accounts for about 74% of the total value of the universe. Since this finding has not yet been explained, its discovery has revealed a profound gap in cosmological knowledge. The relative extent of cosmological knowledge was turned on its head, as it were, by this discovery. Whereas it used to be assumed that the essential elements of matter and force in the universe were understood, what was previously known about them now represents a small island in a largely puzzling landscape of cosmic phenomena.

The details of the history leading up to this breakthrough will not be discussed here. In order to demonstrate the cumulative nature of the phenomena associated with the discovery of the accelerated expansion of the universe, it is sufficient to cite some of the cognitive achievements through which ever more elements of changeability have been verified.[19] Evidence of the changeability of the cosmos in modern Europe goes back to the sixteenth century, when Tycho Brahe interpreted a supernova as a new star; in the nineteenth century, it was inferred from a law of thermodynamics that the universe may be developing toward a completely unstructured state—its so-called heat death; the expansion of the universe was demonstrated in the first two decades of the past century; and its accelerated character was discovered in 1998 through observations of the movement of exploding stars (Type Ia supernovae). Since then, this discovery has been confirmed by other phenomena.[20] Although the cause of the accelerated expansion is not known, that it is a fact is virtually beyond doubt. The data show that the rate of acceleration has been constant for billions of years. Assuming that the rate of acceleration also remains constant in future, then under certain theoretical presuppositions, other world models can be excluded, the most important being the gravitational collapse still regarded as probable in the last century, assuming a future decrease in the acceleration (the Big Crunch), and the future rupture of all bodies postulated just a few years ago on the assumption of an increase in acceleration (the Big Rip).

---

[18]　The significance of the discovery of accelerated expansion was acknowledged with the "Breakthrough of the Year" prize by the American Association for the Advancement of Science's journal *Science* in 1998 and with the Nobel Prize in 2011.

[19]　The cumulative character would become clearer by combining the two stories of the discovery of emptiness, which are separated here for presentation purposes. The discovery of the accelerated expansion presupposes the discovery of distant galaxies and thus a large part of the history of astronomical measurements of distance. The connection is made evident by the relationship between Hubble's measurements of the distances between galaxies and his contribution to the discovery of expansion.

[20]　The phenomena in question are the anisotropy of the cosmic background radiation and studies of the numerical density of galaxy clusters [35] (p. 455), [36].

### 3.2. The Future of Accelerated Expansion

The accelerated expansion of the universe is essentially an expansion of cosmic emptiness, because it enlarges space itself. The change in the cognitive conditions of cosmological knowledge that it will bring about and the dissolution of all bodies to which it will ultimately lead presuppose that it will remain constant into the distant future. The unchangeability of expansion for distant future times has merely the status of a hypothetical assumption based on current measurements, since the latter can only draw on vanishingly small periods of time by comparison with the timescale of the predicted developments. A better assessment of the certainty of the predictions could probably be made if the cause of the accelerated expansion were known.

### 3.2.1. The Insurmountable and the False Emptiness

In roughly 20 billion ($2 \times 10^{10}$) years, the galaxies of the so-called Local Group, to which the Milky Way and the Andromeda Galaxy belong, will coalesce into a single galaxy due to their gravitational attraction.[21] As Lawrence M. Krauss and Robert J. Scherrer have shown, this galaxy will be almost completely cut off from the rest of the universe in about 100 billion ($10^{11}$) years [38,39]. The separation will take place essentially through two mechanisms: On the one hand, in the course of the accelerated expansion, the distance from more and more galaxies outside the Local Group will increase at superluminal velocity, so that ultimately, there will be no causal interaction between other galaxies and the Local Group. On the other hand, the background radiation, which according to the current standard model of cosmology, testifies to the origin of the universe and its accelerated expansion, will probably exercise hardly any effects any longer. It will become more long-wave and will therefore be more likely to be absorbed by the interstellar gas and have only weak intensity [38] (p. 39).

Without sufficient interaction with the rest of the universe, its expansion would no longer be knowable. The relevance of this objective limitation of knowledge for natural philosophy is a function of the possible existence of life forms in the time following this limitation: it is predicted that the end of the stars, and thus the end of the possibility of life forms similar to organic life on Earth, will occur in 100 trillion ($10^{14}$) years. If civilizations existed in the Local Group from our era until the end of the stars, therefore, they would only have knowledge of the expansion of the universe based on their astronomical observations alone for about one-thousandth of that time.

It is conceivable that civilizations that were sufficiently isolated from the rest of the universe would mistakenly regard their own galaxy as an island enclosed by an infinite universe devoid of bodies. At the same time, they would probably overestimate the effect of gravity: without knowledge of the expansion of the universe, they might regard the gravity of the Local Group as sufficient to eventually bring about the end of the universe through a collapse [38] (pp. 40–41). From the perspective of present-day knowledge, this prediction is mistaken, because the accelerated expansion of the universe, assuming it remains constant, is incompatible with its gravitational collapse. Future civilizations would not be able to understand why their predictions did not come true.

It is not only possible future civilizations of the Local Group that would be affected by the prediction of the objective limitations and misdirection of the mind. Since the accelerated expansion affects space uniformly, the cosmic emptiness will create the same false impression throughout the universe of galaxies surrounded by an infinite expanse. This prediction of a universal impairment of cosmological knowledge is without precedent in the history of science and natural philosophy. It is tantamount to the prediction of the inaccessibility of truth. Not least, this prospect also relativizes the current basis of cosmological knowledge, since it could also be that present-day cosmology lacks access to observations that disprove existing theories.

---

[21] For a relevant account, see Nagamine and Loeb [37].

3.2.2. The Completed Emptiness

The initial discussion of the prediction of accelerated expansion led us into future periods of time of between 20 billion and 100 trillion ($10^{14}$) years from now. The last period is around seven thousand times longer than the current age of the universe. Its end will be marked by the expected burning out of the last stars. The production of new stars is already now on the decline. In future, more and more stars will burn out as new stars are created [40] (p. 30). Without stars, all planetary life forms will come to an end. Provided that the expansion of the universe remains constant, the remaining bodies in the universe will then dissolve further into their constituents and in part be dispersed in space and in part collect in black holes.

This assumed development is an extremely intricate process, unfolding at an ever slower rate, of different forms of dissolution (of the various remainders of stars, types of galaxies, and clusters of galaxies) and of temporary formations of structures.[22] In its final and by far the longest phase (between $10^{40}$ and $10^{100}$ years from now or later), there will probably exist only black holes and elementary constituents of bodies dispersed throughout space [41] (pp. 107 ff.). The processes of disaggregation and solidification need not be described in detail here, because only the general tendency is of interest: finally, the black holes will have evaporated[23] and there will be only weakly interacting particles distributed in space at incredible distances corresponding to the diameter of the present universe, at the lowest temperatures or longest-wave electromagnetic radiation (the Big Freeze) [41] (pp. 153 ff.).[24]

When this final state of the accelerated expanding universe occurs depends on whether the protons, which together with the neutrons form the nuclei of atoms, decay or not. If the protons were to decay, which is currently thought to be unlikely, the destruction of physical structure in the universe would occur more quickly than with stable protons. Estimates of how long it will take until bodies dissolve in their entirety and emptiness is completely realized lie with proton decay, in the region of $10^{100}$ years (ibid.), and without proton decay, approximately between the already unimaginably large numbers of $10^{(10^{26})}$ and $10^{(10^{76})}$ years [42] (p. 653), [44] (p. 453).[25]

If someone were to enter this universe in its final state, they would find a vacuum in which it would be extremely difficult, if not impossible, to find any trace of the preceding development of the universe. Together with all material structure, the completed emptiness will presumably also erase all signs of the past. In this scenario, matter will be distributed more and more homogeneously in space, the temperature differences in space will undergo a progressive decrease, and events will occur increasingly rarely, so that the state of the universe will differ less and less from absolute stasis. [41] (pp. 161 ff.). If the burning out of the stars was still an event in the universe, events in the final state itself will come to an almost complete standstill. The probability that this final state will continue indefinitely is very high. It cannot be ruled out that it will exhibit instabilities that could lead to the creation of a new universe [41] (pp. 168 ff.). However, even this universe would have no knowledge of the evolution that preceded the emptiness. The coming emptiness will be completed in the presumed future of the accelerated expanding universe.

There is a similar relationship between the burning out of the last stars and the final state of the accelerated universe as between the two different forms of the end of the world discussed by Immanuel

---

[22]　The authoritative account is still Adams and Laughlin [41].

[23]　Under the conditions of accelerated expansion, the complete evaporation of black holes is considered to be probable [41] (pp. 150 ff.), [42] (p. 650), [44] (p. 451).

[24]　This state is similar to the so-called heat death, cf. Section 3.1.

[25]　It is impossible to provide analogies to other time periods or spatial relationships of cosmological objects for the temporal relations in question. To provide at least an indirect illustration of the unimaginable objective reference of the number $10^{(10^{76})}$, one can try to illustrate its unrepresentability without exponential notation. It is a number with $10^{76}$ zeros. If humankind were to do nothing else from now on except write one zero per person per second, it would only be able to complete (assuming a constant human population) a vanishing fraction of these zeros within the time it will take for the stars to extinguish (approximately $10^{32}$ zeros). In order to write them all down, a further $10^{44}$ human populations doing nothing else would be required. This is $10^{22}$ times larger than the presumed number of stars in the universe ($10^{22}$).

Kant in his essay "The End of All Things". What he calls the "natural end" consists in the perfectly imaginable complete destruction of physical orders before the last day. What he calls the "mystical end" is the humanly completely incomprehensible end of all change, and thus the end of time, which at some point follows that destruction: "But that at some point a time will arrive in which all alteration (and with it, time itself) ceases—this is a representation which outrages the imagination. For then the whole of nature will be rigid and as it were petrified." [45] (pp. 227 f.).[26] What for Kant were still mere ideas that "lie wholly beyond our field of vision" [45] (p. 225)—that is, the natural and mystical end—become probable scenarios in modern cosmology: with the end of the stars, the physical orders will in all probability cease to exist; in the final state of the universe, the difference between change and stasis will become arbitrarily small. Nature will still exist, but it will be scattered in infinite spaces and will be "rigid and as it were petrified".

*3.3. The Proximity of the Distant Emptiness*

The periods during which the cosmic expansion exercises its effects on the conditions of knowledge and on the end of bodies are incommensurate with the horizon of everyday experience. They cannot be represented intuitively through analogies to spatial distances. It is all the more surprising that thematizations of the still-distant cosmic conditions in popular presentations have met with a broad interest.[27] They are objects of media attention that are not concerned with practical relevance, but with entertainment, education, and/or world orientation. Natural philosophy asks: How does everyday lifeworld experience, which nonspecialists can be assumed to possess, manage to share in cosmological knowledge? The thesis defended here is that the lifeworld imagination is able to bridge great temporal differences with relative ease. This assumption presupposes that the temporal horizon of the lifeworld and the cosmological temporal horizon have gradually drifted apart in modern times [15]. While the limits of what is cosmologically predictable lie further and further away from the present, the everyday practical stance has remained essentially concentrated on the present. The result is not only, as is generally assumed, a lifeworld-based weakness when it comes to perceiving the relevance of the future,[28] but also the opposite lifeworld deficiency in the ability to differentiate between orders of magnitude that are far removed from the present. Thus, what is temporally very distant can be shifted into the horizon of what is near in time.[29] Very distant events, such as the emptiness of the final state of an expanding universe, can therefore acquire increased significance despite their temporal distance.

But how should we conceive of the completion of emptiness? Does not thinking itself become empty when, paradoxically, it tries to imagine a condition of almost perfect changelessness and timelessness? How can it conceive of all products of life and thought being destroyed in the coming final state?

## 4. Coming Emptiness

I see the twofold sense in which the cosmic emptiness is coming as an expression of its objectivity.[30] I have described emptiness as coming, because knowledge of it has come to human beings, as it were

---

[26] Sten Odenwald describes the difference between the burning out of the last stars and the final state of the accelerated universe as two different forms of death: "the death of the living biosphere, the death of the cosmos" [46] (p. 155). Paul C. Davies speaks more correctly of the eternally dying universe [47] (pp. 83–100).

[27] The topics of the expanding universe are very much present in the science pages of newspapers, on TV science programs, and in popular science magazines. See also the popular writings of physicists mentioned in Footnote 7 and [6,41–43,46,47].

[28] Thomä [48] and Großheim [49] are exemplary examples of this complaint, which does not criticize the lack of awareness of future cosmic events, but of events extending to the next generations at most.

[29] This assumption is based on the study of the temporal structures of the lifeworld in Schütz and Luckmann [50] (pp. 73 ff.); cf. Schiemann [4] (pp. 115 f.).

[30] "Objectivity {XE "Objectivity"}" does not only mean the independence of knowledge from individual factors such as attitudes, opinions, or convictions; it also denotes the property of facts that they are not susceptible to the effects of human action in an epoch- and cross-cultural sense, without thereby being historically unchangeable.

(as discussed in Sections 2.1 and 3.1), and its relevance in the cosmos is likely destined to increase (as discussed in Sections 2.2 and 3.2). The longer and more closely emptiness has been studied since the beginning of modernity, the larger became the spaces over which it was found to extend and the more reliable the knowledge of its nature became.[31] The impressive extent of the current emptiness of the universe is demonstrated by its average (baryonic) matter density of $10^{-30}$ g/cm$^3$, which is around 30 orders of magnitude lower than the average density of the bodies of stellar systems.[32] With the discovery of the accelerated expansion of the universe, emptiness acquires the status of an increasingly characteristic feature of the cosmos. If the value of the acceleration remains unchanged, the future of the cosmos is fixed: emptiness will continue to spread until it ultimately engulfs the entire universe. According to this, the universe is on a deterministic path to emptiness. For all the impressive range of this cosmological prediction, its hypothetical character must still be taken into account, which increases with the magnitude of the predicted time frames.

For the natural philosophical view concerned to establish the basic features of nature in its relationship to human beings, the fact that the determined future of emptiness contrasts with its indeterminate present is of interest. Compared to the previous cumulative increase in knowledge of the emptiness and its predicted distance dominance, the developmental trends in current research in the closer cosmic environment of emptiness are rather indeterminate. The results of exoplanet research on interstellar space, which has just begun, remain completely open. Is the Earth surrounded by habitable planets, some of which may even have life on them? Will we soon discover signs of extraterrestrial civilizations through observations of the atmospheres of habitable planets? Or will the observational search for extraterrestrial life remain unsuccessful for the time being? Will long and risky journeys by unmanned probes be necessary to gain reliable knowledge about planets in other solar systems?

The punchline of the natural philosophical reflections on the near future of the relevance of the cosmic emptiness for natural philosophy in the context of exoplanet research is that this relevance will probably increase, whatever the result of the research turns out to be. With the discovery of extraterrestrial life, interstellar destinations would adopt a concrete form that would first make the problems of reaching them painfully apparent. The permanent lack of proof of the existence of extraterrestrial life could be an indication of the rarity and loneliness of human life. The hostility to life of the cosmos would extend not only to the emptiness, but also to the other planets.

The problems of the reachability of planets of neighboring suns are primarily a function of the amounts of energy required for manned space flights or, alternatively, of the requisite travel times. If humans would not be in the position to provide a space flight with amounts of energy several times the current annual energy consumption of the Earth, they would have to countenance travel times many times longer than a human lifetime. Against the background of this dilemma, it seems plausible to assume that scope of human actions in the cosmos remains tied to the Earth for the time being. The limited scope of human action contrasts with the increasingly far-reaching and precise observational possibilities.

In addition to the findings mentioned in this section, the results of the natural philosophical study of emptiness include the justification of the expanded concept of natural philosophy obtained through the reflection of cosmological knowledge (see Section 1), of the categorical difference between interplanetary and interstellar distances (see Section 2.2), of the dependence of future cosmological conditions of knowledge on the accelerated expansion of the universe (see Section 3.2.1), and of the

---

[31] The historical process could be formulated trenchantly as follows: "The closer a star was looked at, the greater the distance from which it looked back", echoing Karl Krauss's saying: "The closer you look at a word, the greater the distance from which it looks back at you" [51] (p. 362). Like words, the stars also lose their initially self-evident closeness and familiarity with increasing knowledge. On this general trend in the history of measurements of cosmic distances, see also Ferguson [16] (p. 180).

[32] The value for the mean density of the universe is generally accepted; see, for example [52] (p. 22). The values for the mean density of the stars and of rocky and gaseous planets are estimated to be between 0.5 and 5 g/cm$^3$.

possibility that cosmological questions can be rendered intelligible in lifeworld terms, despite the fact that their spatial and temporal dimensions far exceed those of the lifeworld (see Section 3.3). On the one hand, the significance of the Earth-bound nature of human existence is being enhanced (see Section 2.2). Presumably, in the long run, humans will not be able to reach Earth-like planets and return to Earth, if they so desire. The emptiness surrounding the Earth means that we must be very careful about how we deal with the conditions of existence of life on Earth. On the other hand, the future coming emptiness relativizes all meanings, not only cosmological ones, since in the final state of the universe devoid of bodies, these meanings will have crumbled to dust, as it were (see Section 3.2.2).

**Funding:** This research received no external funding.

**Acknowledgments:** I would like to thank Ciaran Cronin for his excellent translation of the German original, two anonymous reviewers for their valuable and stimulating comments, and Niklas König for his research on numerous points, his editorial work, and his openness for discussion.

**Conflicts of Interest:** The author declares no conflict of interest.

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
