# Peer review of "The Coming Emptiness: On the Meaning of the Emptiness of the Universe in Natural Philosophy"

_philosophies, doi:10.3390/philosophies4010001_

Round 1

Reviewer 1 Report

Section 3.1, lines 292-324: The Author takes the accelerating expansion of the universe as an established fact.

COMMENT:

The present Reviewer cannot agree with this. My analysis of the work by the two groups led by Perlmutter and by Riess and Schmidt, respectively, indicates that they have made a significant methodological error in the measurements of the distances to the supernovae Ia. Later measurements based on the temperature variations in the CMBR and distances between clusters make the same error. In my opinion, there is no accelerated cosmic expansion. Irrespective of whether the Author and the scientific establishment are right or I am, it is still of interest to see the consequences in the philosophy of nature of the mainstream view.

Isn't the Big Rip in line 324 the same as the dissolution of all bodies? In lines 320-324, the Author writes that under certain circumstances, there can be no Big Rip, while in other places, he states that a future dissolution of all bodies is unavoidable. This is a contradiction. The truth is that the acceleration of the expansion of the universe is not in itself sufficient to imply a dissolution of all bodies. Further speculative hypotheses must be added to get either of the alternatives.

Page 10, footnote 22: "Under the conditions of accelerated expansion, evaporation of black holes is considered to be probable."

COMMENT:

The evaporation is caused by the Hawking radiation. As far as I have understood, this kind of radiation is independent of whether the expansion is accelerating or not.

Page 11, lines 401-402: "Together with all material structure, the completed emptiness will also erase all signs of the past."

COMMENT:

According to a theorem in QM, information is indestructible.

Page 11, lines 408-409: "It cannot be ruled out that it will exhibit instabilities that could lead to the creation of a new universe."

COMMENT:

Work by Andrei Linde shows that the inflation theory implies that the creation of new universes is virtually certain.

Page 13, lines 468-469: "Will long and dangerous journeys by unmanned probes be necessary to gain reliable knowledge about planets in other systems?"

COMMENT:

How can unmanned journeys be dangerous?

IN GENERAL: This is a well written article well worth publication. 

Reviewer 2 Report

See attached file.
